# The Danish SoL Project: Effects of a Multi-Component Community-Based Health Promotion Intervention on Prevention of Overweight among 3–8-Year-Old Children

**DOI:** 10.3390/ijerph18168419

**Published:** 2021-08-09

**Authors:** Tine Buch-Andersen, Frank Eriksson, Paul Bloch, Charlotte Glümer, Bent Egberg Mikkelsen, Ulla Toft

**Affiliations:** 1Center for Clinical Research and Prevention, Bispebjerg and Frederiksberg Hospital, Nordre Fasanvej 57, 2000 Frederiksberg, Denmark; ulla.toft@regionh.dk; 2Section of Biostatistics, Department of Public Health, Copenhagen University, Øster Farimagsgade 5, 1014 Copenhagen, Denmark; eriksson@sund.ku.dk; 3Health Promotion Research, Steno Diabetes Center Copenhagen, Niels Steensens Vej 2, 2820 Gentofte, Denmark; paul.bloch@regionh.dk; 4Center for Diabetes, Copenhagen Municipality, Vesterbrogade 121, 1620 Copenhagen, Denmark; FS0H@suf.kk.dk; 5Department of Geosciences and Natural Resource Management, Copenhagen University, Rolighedsvej 23, 1958 Frederiksberg, Denmark; bemi@ign.ku.dk

**Keywords:** BMI, anthropometry, children, health promotion, multi-component, community, intervention study, Denmark, supersetting

## Abstract

The aim of the study was to determine the effects of a multi-component community-based health promotion intervention on body mass index (BMI) z-scores and waist circumference (WC) in three- to eight-year-old children. A quasi-experimental design was adopted to evaluate the effects of the SoL intervention involving three intervention and three control communities. The 19-month intervention was based on the supersetting approach and was designed to promote healthier eating and physical activity among children and their families. BMI z-scores and WC were measured at baseline and follow-up. At baseline, 238 (54%) and 214 (51%) of all eligible children were measured from intervention and control, respectively. The change over time in the BMI z-scores of children from the intervention group was significantly different from that of the control group (*p* = 0.001). BMI z-scores increased over time in the intervention group in contrast to the control group, whose BMI z-scores decreased (difference in change between groups 0.19 z-scores 95% CI 0.08, 0.30). No significant differences were observed for WC. The results showed no favourable effects of the intervention of Project SoL on BMI z-scores and WC in children. Further studies based on a larger sample size and a longer intervention duration are needed.

## 1. Introduction

During the past half-century, most parts of the world have seen a shift in disease patterns from communicable diseases to chronic non-communicable diseases (NCDs) and obesity. In recognition of this change, there has also been a shift in how we understand health and how we organise and implement health programmes. In 1986, the WHO Ottawa Charter noted that: “Health is created and lived by people within the settings of their everyday life; where they learn, work, play and love” [1]. Since then, the concern about public health has increased, and it is now clear that the development of obesity and NCDs are rooted in social, environmental and behavioural determinants, including education, nutrition, physical activity and social relations. Health behaviours are shaped by influences at multiple levels (personal, organisational, environmental and policy) and are often founded in early life [2,3]. Early life primary prevention is, therefore, important.

Numerous studies have been conducted in the area of obesity prevention, and despite large, well-designed studies, most of them fail to show long-lasting and sustainable effects [4].

A community-based approach offers great potential for health promotion among children and families. It can be designed as a complex intervention, including multiple intervention levels, components, stakeholders and settings to effectively change lifestyle behaviours [4,5]. If including capacity-building elements and structural changes, it has the potential to be sustainable.

Publications reporting the effects of community-based studies with a complex intervention design using community engagement and capacity-building processes and aiming at preventing childhood overweight are scarce [6,7]. These studies mainly involved schools as the dominant setting, were mostly conducted in American and Australian contexts and mostly involved populations with a high prevalence of overweight. The Danish Project SoL (from the Danish Sundhed og Lokalsamfund/Health and Local Community) sought to fill this gap by implementing a multi-component and integrated health promotion intervention in selected local communities based on the principles of the supersetting approach [8,9,10]. It was the first Danish and Nordic study of this kind. The project aimed to promote healthier lifestyles among Danish children aged 3–8 years and their families. There was an emphasis on mobilising community resources, strengthening social networks and promoting healthier eating and physical activity. A secondary objective was to prevent overweight among children. Preliminary results of Project SoL showed favourable effects of the intervention on sales of fish, whole grains and vegetables compared with the control group [11]. Further results regarding supermarket sales, awareness and measurements of behaviours are yet to be published.

The purpose of the present study was to examine the effects of Project SoL on anthropometric measures related to overweight in three- to eight-year-old children.

## 2. Materials and Methods

### 2.1. Intervention Design

The conceptual framework as well as the intervention design of Project SoL have previously been described in detail [8,9,10]. In short, the intervention of Project SoL was conceptually rooted in the supersetting approach and implemented using action research methodology [8]. The supersetting approach is characterised by being integrated but also participatory, empowering, context-sensitive and knowledge-based. A program theory of the project was developed to guide both the development of interventions and the evaluation [9]. In accordance with the supersetting approach, the interventions in Project SoL were implemented in a coordinated and integrated manner in several everyday life settings to promote intensity, impact, synergy and sustainability.

The project had a total duration of four years, including a 19-month intervention period. The intervention targeted young children and their families and aimed at promoting physical activity and healthy eating. The intervention was setting-based and included childcare centres, primary schools, supermarkets, local mass media and local communities. The intervention components included participatory learning methods (Future Workshop Scenario, Mosaic Method), educational activities, environmental strategies, press coverage, health campaigns, social media, local community activities and public events. Interventions were not predetermined but developed jointly with citizens and professional stakeholders as the project unfolded [9]. Focus on child weight status was minimised in the development of interventions because of a past intervention that was considered to cause stigmatisation due to weight and obesity.

The implementation of activities in Project SoL involved several local stakeholders within the municipality, primary schools, after-school centres, childcare centres, supermarkets, media and a number of civil society organisations and resource persons with expertise in nutrition, cooking, recreation and physical activity. An executive committee consisting of the local coordinator in addition to three senior researchers were in charge of the day-to-day coordination and planning. Common overall themes were addressed across settings and communities (e.g., “taste and senses”, “fruit and vegetables”, “nature and movement”) to ensure coordination and integration of activities. Some activities were carried out across settings, and activities varied from setting to setting and from community to community, depending on the local ideas and motivation. An overview of intervention themes and activities can be found here [9].

### 2.2. Study Design

The evaluation design of Project SoL was quasi-experimental. Intervention and control sites were located in two different and geographically separate municipalities of Denmark. They were selected based on a number of similarities in the number of childcare centres, primary schools and supermarkets, as well as socio-demographic characteristics and the prevalence of non-communicable diseases. Both municipalities had high proportions of citizens with a low socio-economic position and a high prevalence of health risk factors and non-communicable diseases [12]. Three communities in the Danish Regional Municipality of Bornholm were selected for intervention, while another three communities in Odsherred Municipality were selected as controls. Four childcare centres and three schools from the intervention communities participated. In the control communities, three childcare centres and three schools participated. Anthropometric data were collected at baseline in September 2012 and at follow-up in April 2014. Several other quantitative and qualitative data were also collected during the project period, including questionnaire data on nutrition, physical activity, cooking, shopping, use of media, use of nature, mental health and social capital. Further details on the evaluation design of Project SoL can be found elsewhere [10].

### 2.3. Subjects

Children from participating primary schools, year zero to two (about six to eight years of age) and childcare centres (about three to six years of age) were invited to participate in measurements of anthropometry. This was the age range that was actively engaged in the intervention.

Parental informed written consent was obtained for all children participating in the measurements. Civil registration numbers (CPRs) were collected from the consent forms and enabled the use of national register data provided by Statistics Denmark. In Denmark, all individuals are assigned a unique 10-digit CPR number at birth or when they receive a permit to stay in Denmark. This number can be linked to central registries with information on socio-demographic characteristics of the study population.

Of the 861 eligible children, 513 (60%) consented to participate. Figure 1 presents the flow of recruitment, participation and analysis. Participation levels varied according to measurement type and year. Children who were enrolled in the childcare centres or schools after baseline could also participate in the measurement but were not included in the analyses nor are they included in the flow diagram.

For some of the children, the CPR number was not available or useful (due to parents’ reluctance to disclose the number on the consent form or due to errors in the stated number), and/or covariate data was missing, which excluded them from the analysis (Figure 1). Three of the anthropometric measurements were considered invalid due to measurement errors, and one child had missing information on school affiliation at baseline (Figure 1).

### 2.4. Anthropometric Measurements

The children were examined barefooted and in light clothing by research staff in the involved childcare centres and schools. Height was measured to the nearest 0.1 cm using a portable stadiometer (Leicester Height Measure), and body weight was measured to the nearest 0.1 kg using a digital scale (Tanita BWB-800). Waist circumference was measured to the nearest 0.5 cm midway between the lowest rib and iliac crest using an ergonomic circumference measuring tape (Seca). It was measured in triplicate, and the mean was calculated and used in the analyses.

BMI was calculated as weight in kilograms divided by height in metres squared and was converted to z-scores using the LMS method [13] with the use of Danish reference data [14]. The International Obesity Task Force age-specific BMI cut-offs were used to classify children’s weight status as underweight (thinness grades 1–3), healthy weight, overweight or obese [15,16].

### 2.5. Socio-Economic and Family Status

Information on family and socio-demographic characteristics of participants was obtained through central registries [17]. The categorisation and calculation of variables was according to the standards of Statistics Denmark and based on standardised procedures [18,19,20]. Details on the categorisation and calculations are provided in Appendix A.

### 2.6. Statistical Analysis

Differences between the intervention and control groups at baseline were statistically determined using an unpaired *t*-test, chi-squared test or Fisher’s exact test. Participant attrition was analysed for intervention and control groups separately, and differences (between children that were followed up and children that dropped out of the study) were tested using an unpaired *t*-test or Fisher’s exact test.

We used a longitudinal linear mixed model (SAS PROC MIXED) to assess differences in anthropometric outcomes between the intervention and control groups [21]. An unstructured residual covariance matrix was used to account for serial correlation of observations from the same individual over time. To account for similarities among children within the same community and school or childcare centre, the model included two random effects: community and school or childcare centre. We included group (intervention/control), visit (baseline/follow-up) and the interaction between group and visit as fixed effects. The test for the group visit interaction compares the intervention and control groups in terms of their patterns of change from baseline [22]. The model was also adjusted for covariates: child sex, child age, parental education, household income and family type. An additional model was run for anthropometric data, which was limited to children who participated at both baseline and follow-up. The mixed model was fitted using maximum likelihood, providing valid inference with incomplete follow-up data under the missing at random (MAR) assumption [23].

The assumptions were checked using residual plots and histograms. The analysis excluded children not enrolled at baseline, but missing follow-up measurements were allowed. Sample sizes differed for each of the analyses because of excluded values for height and missing data for baseline waist circumference (Figure 1).

Statistical analyses were performed using the SAS statistical software package version 9.4 (SAS Institute Inc., Cary, NC, USA).

## 3. Results

At baseline, 238 children (54%) participated from the intervention group, and 214 (51%) participated from the control group (Figure 1). A total of 175 children (74% of those participating at baseline) from the intervention group also participated at the final follow-up. In the control group, 170 (79% of those participating at baseline) also attended the final follow-up.

Table 1 summarises the children’s characteristics at baseline for the intervention group and the control group. The distribution of parental education levels differed significantly between the intervention group and the control group (*p* < 0.001). The other baseline characteristics did not differ. A participant attrition analysis was performed (data not shown), which detected a few differences. In the intervention group, children who dropped out (prior to follow-up) were significantly younger than those who were followed up (−0.44 years, 95% CI −0.87 to −0.04, *p* = 0.04). In addition, the distribution of family status categories differed significantly between children who were followed up and those who dropped out (*p* = 0.03). Single parents dropped out more than couples. Otherwise, none of the characteristics differed significantly between those who were followed up and those who were lost to follow-up for either the intervention group or the control.

Table 2 shows baseline and follow-up summary statistics. Table 3 shows the adjusted differences between the intervention group and the control group. The prevalence of overweight and obesity decreased by 1 percentage point from 15% at baseline to 14% at follow-up in the intervention group and by 3 percentage points from 12% to 9% in the control group (Table 2). There were significant differences in anthropometric measures from baseline to follow-up between the intervention group and the control group (Table 3). BMI z-scores increased by a mean of 0.09 (95% CI 0.01 to 0.17) z-scores in the intervention group, in contrast to the control group whose BMI z-scores decreased by 0.10 (95% CI −0.18 to −0.02) z-scores, a difference in change between groups of 0.19 (95% CI 0.08 to 0.30) z-scores (*p* = 0.001). There was no significant difference for waist circumference (Table 3). Analyses of intervention effects were also performed for children who participated at both baseline and follow-up (*n* = 283 for BMI z-scores and 280 for waist). Results were similar; the adjusted difference for BMI z-scores was 0.20 (95% CI 0.08, 0.31, *p* = 0.0009), and there was no significant difference for waist circumference.

## 4. Discussion

The aim of this paper was to investigate the effect of a multi-component community-based health promotion intervention on BMI z-scores and WC in three- to eight-year-old children. The results showed no beneficial effects on BMI z-scores and WC at 19-month follow-up in the intervention group compared with the control group. A small but proportionally larger decrease in BMI z-score was observed in the control group compared with the intervention group. Additionally, the results showed a decreased prevalence of overweight and obesity by 1 percentage point in the intervention group and by 3 percentage points in the control group. The results of the SoL project regarding weight-related measures must be interpreted in a broader context that considers a number of circumstances and limitations of the study.

The results of this study are in line with research from a large quantity of RCTs [4] aiming to prevent overweight and obesity in children. Most did not result in a reduction in BMI z-score or other measures of overweight and obesity. Results from community-based studies with an approach similar to the SoL study [6,7,24,25,26] are sparse and mixed. Some of these studies have shown favourable effects on one or more indicators of childhood overweight or obesity in all or subgroups of the intervention group. The sample sizes were often considerably larger (>1000) than the sample size of Project SoL, which increased the chance of observing small effects and enabled them to differentiate between different subgroups. Some of the studies included policy changes and structural changes (e.g., healthier school catering in the Shape Up Sommerville project [27]) to a greater extent than Project SoL which may also explain the variation in effects. The intervention duration of community-based studies varies, but, in general, a long project duration is recommended [28] because of the lower intensity of intervention activities. Project SoL was based on the supersetting approach and used an action research methodology. Using this approach, the project emphasised trust building between partners and continuously involved local stakeholders and citizens in planning intervention activities. This meant that the initial intensity of intervention activities was rather low, especially during the first year of implementation. It is therefore possible that the intervention duration of Project SoL was too short for measuring impacts. A minimum of 3 years for implementation has been recommended [28], and in the French part of the EPODE study, it took more than 10 years before they observed an effect of the intervention on overweight prevalence [29].

The present study observed a significant group difference (0.19 z-scores) and an unanticipated decrease in BMI in the control group. The reasons for this are unclear. Because the decrease in BMI z-scores in the control group was very modest and because of the methodological limitations of the study, we think that we should be careful when interpreting the effect sizes. It is likely that the sample size was too small, and when conducted in a non-randomised design, it is more difficult to take into account differences between intervention and control. According to the national database, Børnedatabasen (in English: The Children’s Database) [30], 3% of the children living in Odsherred were obese in 2012 and 2013 (total sample *n* = 265 and 278), which was not reflected in our data. The number of children included in the database is small, but the possibility exists that the anthropometric data of the present study suffered from limited representativeness. We have no reason to believe that the intervention caused disbenefit or harm to the study population. All project activities were developed jointly by stakeholders in the local community and presented to the study population as free and unconditional offers they could accept or decline.

### 4.1. Implications

There is an urgent need to build an evidence base of large-scale community-based studies conducted in Denmark and Scandinavia to prevent overweight and to promote wellbeing and health among children and their families. Based on the experiences of Project SoL, the intervention duration should be longer, and the number of intervention activities should be higher and more frequent. Based on results from other studies, it is likely that a higher degree of policy and structural changes in the communities could form a better basis for improvements in health. In terms of evaluation, future studies based on the supersetting approach should include a large sample size (preferably thousands from several communities) and plan for a minimum of 3 years of follow-up. If sufficient power, the evaluation of anthropometric measures should include analysis of subgroups, e.g., normal weight vs. overweight and socio-economic groups. We also recommend for future studies to analyse outcomes by setting. Further recommendations and lessons learned are listed in previous publications from the SoL research group [9,10].

### 4.2. Strengths and Limitations

One of the strengths of the present study was its strong conceptual foundation and design. It was conceptually rooted in the supersetting approach [8] and used a controlled evaluation design. National reference data was used for the generation of BMI z-scores using the LMS method, and the use of national register data enabled the use of relevant covariates that were not available from questionnaires. National register data provide reliable data with a high degree of completeness. Therefore, we had accurate data on socio-economic and family status without recall bias for all children with a CPR number.

The present study had limitations. At baseline, the participation rate was approximately 50%, which is not lower than that observed in other studies, but there is a risk of selection bias if the children measured differ from the children who were not measured. As previously mentioned, it is possible that the control group was not representative in relation to the prevalence of obesity. Project SoL had no information on non-participants, but we analysed participant attrition (participants who did not participate in follow-up measurements). In the intervention group, children of single parents dropped out significantly more than children of couples. The attrition among this group can perhaps be explained by factors related to time constraints. According to the literature, parental education is the strongest and most consistent dimension of SES associated with overweight and obesity in children in Western developed countries [31]. Fortunately, for this variable, no difference was observed between follow-up and dropout children. Potential bias related to loss to follow-up was minimised, as the statistical methods we applied are unbiased if the data are MAR [23]. Due to practical and economic resource limitations, different data collectors were employed to measure height, weight and WC, which means that inter-data collector variability may have occurred. All data collectors were, however, trained and supervised by an experienced nutritionist to standardise measurements as much as possible. Measuring waist circumference requires more training and is more prone to measurement errors [32], but we chose to collect this measurement, as it may add additional information besides BMI.

## 5. Conclusions

The results showed no favourable effects of the SoL intervention on BMI z-scores and waist circumference in children compared with the control group. There is a need for larger and more long-term intervention studies that build on the supersetting approach and are designed to influence and evaluate childhood overweight outcomes.

## Figures and Tables

**Figure 1 ijerph-18-08419-f001:**
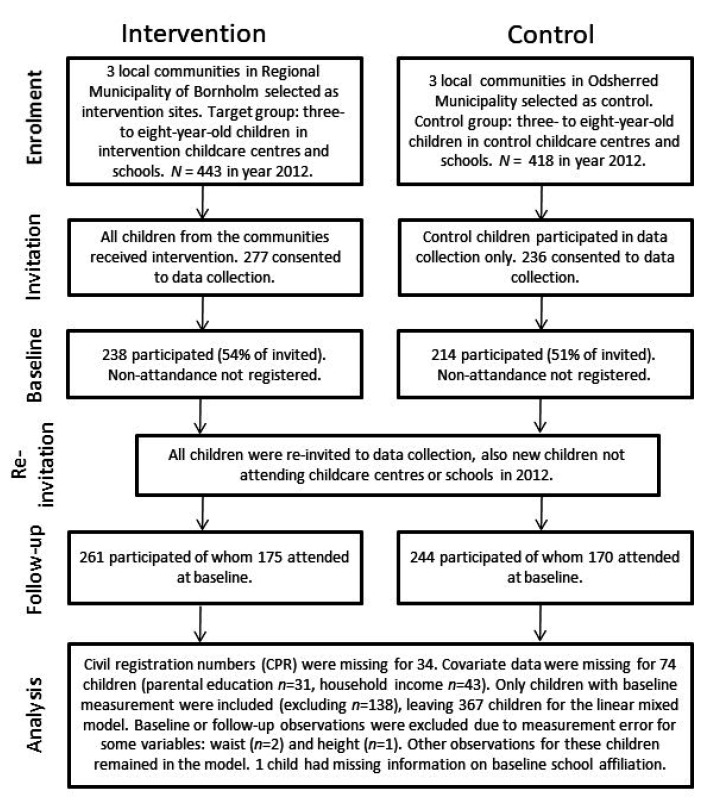
Flow diagram of recruitment and analysis.

**Table 1 ijerph-18-08419-t001:** Child baseline characteristics.

	Intervention Group (*n* = 201)	Control Group (*n* = 166)
	Mean or Frequency	Mean or Frequency
Age, years (SD)	6.51 (1.49)	6.21 (1.80)
Female (%)	105 (52)	90 (54)
Height, cm (SD) ^a^	120.76 (10.84)	118.58 (11.92)
Weight, kg (SD)	23.72 (5.44)	23.04 (5.33)
Parental education ***		
Primary/secondary school (%)	23 (11)	44 (27)
Vocational education (%)	100 (50)	65 (39)
Academy or bachelor (%)	71 (35)	45 (27)
Master or PhD (%)	7 (4)	12 (7)
Household income		
Lower quartile (%)	38 (19)	42 (25)
Lower middle quartile (%)	86 (43)	57 (34)
Upper middle quartile (%)	49 (24)	44 (27)
Upper quartile (%)	28 (14)	23 (14)
Family status		
Married couples (%)	125 (62)	111 (66)
Registered partnership (%)	1 (1)	0 (0)
Couple living in consensual union (%)	43 (21)	26 (16)
Cohabiting couples (%)	13 (7)	11 (7)
Single parent (%)	19 (10)	19 (12)

Abbreviations: SD, standard deviation. ^a^
*n* = 200 in the intervention group due to measurement error of height. *** *p* < 0.001.

**Table 2 ijerph-18-08419-t002:** Baseline and follow-up summary statistics for anthropometric measures.

	Intervention Group						Control Group					
	Baseline			Follow-Up			Baseline			Follow-Up		
	*n*	Median	P25, P75	*n*	Median	P25, P75	*n*	Median	P25, P75	*n*	Median	P25, P75
BMI z-score ^a^	200	0.15	−0.58, 0.84	151	0.04	−0.61, 0.91	166	0.32	−0.32, 0.77	130	0.15	−0.35, 0.85
Waist circumference, cm	201	54.67	51.33, 57.17	150	56.58	53.17, 60.33	165	53.33	51.00, 56.33	129	56.17	52.83, 59.00
Weight class ^b^	*n*	%		*n*	%		*n*	%		*n*	%	
Underweight	12	6		10	7		8	5		9	7	
Healthy weight	160	80		120	80		138	83		110	85	
Overweight	25	13		17	11		20	12		11	9	
Obese	3	2		4	3		0	0		0	0	

Abbreviations: BMI z-score, standardised body mass index; P25, 25th percentile; P75, 75th percentile. ^a^ BMI z-score calculated against the 2014 Danish growth reference. ^b^ Weight classes as defined by the IOTF. Underweight includes thinness grades 1–3.

**Table 3 ijerph-18-08419-t003:** Adjusted changes (95% confidence interval) for intervention and control and group difference ^a^.

	Intervention Group	Control Group	Group Difference ^a^		
	Adjusted Change (CI) ^b^	Adjusted Change (CI) ^b^	*n* (Child/Obs)	Adjusted Change (CI) ^c^	*p*
BMI z-score ^d^	0.09 (0.01, 0.17)	−0.10 (−0.18, −0.02)	367/646	0.19 (0.08, 0.30)	0.0013
Waist circumference, cm	0.29 (−0.31, 0.90)	−0.09 (−0.71, 0.53)	367/644	0.38 (−0.26, 1.02)	0.2374

Abbreviations: BMI z-score, standardised body mass index; CI, confidence interval. ^a^ Analysed by a longitudinal linear mixed model adjusted for age, sex (waist only), parental education, household income and family status. ^b^ Difference between baseline and follow-up (follow-up minus baseline) for intervention and control groups adjusted for covariates. ^c^ Difference in change (follow-up minus baseline) between intervention and control groups. ^d^ BMI z-score calculated against the 2014 Danish growth reference.

## Data Availability

The data presented in this study are available on request from the corresponding author. The data are not publicly available according to the Danish Act on Processing of Personal Data.

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
