# Peer review of "The Danish SoL Project: Effects of a Multi-Component Community-Based Health Promotion Intervention on Prevention of Overweight among 3–8-Year-Old Children"

_ijerph, 2021, doi:10.3390/ijerph18168419_

Round 1
Reviewer 1 Report
Thank you for making the edits, I believe all the edits make the paper stronger. I only have 3 points (in priority order):
- I really like your text about being cautious in interpreting effect sizes. I would strongly recommend deleting: ."....we believe that a cautious interpretation of the study findings is that the intervention had 245
“no effects” on BMI and WC". I would no interpret but leave this to the reader. - please reference the following "Focus on child weight 93
status was minimized in shaping interventions because of a past intervention that was 94
considered to cause stigmatization due to weight and obesity". and perhaps define what you mean by 'shaping interventions' or reference the term. - comment: readers might wonder why this publication comes 7 years after data collection so maybe put a sentence in?
Reviewer 2 Report
The quality of the manuscript has increased significantly as a result of the revision. Above all, the introduction has become clearer through the inclusion of the determinants, a model and also the holistic understanding of health. In addition, the structure of the discussion is much better.
Nevertheless, my great concern remains that the goals of the intervention (PA, nutrition) do not match ideally the outcome of obesity, especially if other determinants of obesity are not taken into account. I already mentioned this in my first review.
You mentioned, that other outcomes, such as PA and nutrition, will be published in later papers. This is in my opinion not the right way. It would make more sense, to publish first these outcomes (PA, nutrition) and then, in another step to publish the data on obesity, and to calculate more complex models, where for example PA and nutrition are involved. I still cannot see, where the gain in knowledge is (what is innovative), when you write several times, that the important outcomes and data will be published in other papers.
In addition, the question of why mental health was not recorded remains more or less unanswered.
Round 2
Reviewer 2 Report
I completely understand your arguments and see, that you are not willing to change the paper fundamentally.
This manuscript is a resubmission of an earlier submission. The following is a list of the peer review reports and author responses from that submission.
Round 1
Reviewer 1 Report
Please see attached.

Reviewer 2 Report
General
Overall, the paper takes up a socially important topic, namely overweight and obesity in children. With regard to the study design, the study is well conducted and the approach of a community-based intervention is also promising. Nevertheless, in my opinion, the paper has a clear weakness: It argues with health behaviors such as physical activity and healthy nutrition, and this is also the aim of the described intervention. The success of this intervention is then only recorded via BMI and WC. In the case of variables, on the one hand, they focus on the physical dimension of health. On the other hand, there are a number of determinants that influence BMI and WC, none of which were taken into account. The question therefore arises as to whether the intervention goals and the measured outcomes fit. Therefore, the following general questions arise for me:
- Why was health only measured with BMI and WC and not also with psychosocial aspects of health?- When it came to increasing PA and improving healthy nutrition, why weren't these two outcomes included?- Why were other energetically determined behaviors, such as sedentariness or media use, not taken into account and included in the study?- It is not at all clear whether the children of the IG participated in any intervention of the program, because that was not asked. - Why was the knowledge (health literacy) in PA and nutrition not measured? Maybe, that would have been a much better outcome, or the actual behavior (PA, nutrition)?
Abstract
- It would be very helpful, to describe why this research is important and what the benefits could be. Therefore, I miss mentioning the relevance of this paper.
- The conclusion is too general and not precise enough.
Introduction
- Why this narrow focus on healthy weight? It is about health in general, is it a holistic understanding of health (physical, psychological and social)?- It would be important to include some kind of determinant model of health, with central aspects such as: What are the main causes of obesity? The behaviors mentioned are important here, such as PA and nutrition. But what are the determinants of these behaviors in a second step?- Health and psychosocial consequences à Which understanding of health is based on, why not using a holistic view?- Better clarify the understanding of health and well-being.- Better clarify the connection between the behavior and the changes in the community-based area. So how is community change related to PA and nutrition? (especially focussing on nutrition)- Work out even better what the innovation of the paper is.
Materials and methods:
- The intervention targeted young children and their families and aimed at promoting physical activity and healthy eating: Then why not using these behaviors as outcomes, but overweight?- 2.3: Is there a systematic difference between participants and non-participants?- 2.5: 99% Danish: is that actually representative? Does it seem like you have mainly reached children without a migration background?
Results:
- Table 1: I do not understand the calculation of the sample size (N)
- Analyses (table 2): Why weren't only those included who took part in both measurement times?
Discussion:
- The discussion is good and very critical, but: Why was overweight taken as one of the central outcomes, if previous studies have not shown anything? You could have anticipated that and taken other outcomes.- You mentioned the results of earlier studies: What could have been done better to get better results?
- ll 235: Which indicators, why not measuring them?
- My main concern: One can therefore establish little or no causality between the intervention and possible effects?- A clearer structure of the discussion would be very helpful for the reader: classification, evaluation, description of causes etc.
- It was prior to the study known, that in Odsherred there are only 3% of obese children, so why implementing a program for the reduction of obesity – why not running this study in a city, where more children are obese?
Conclusion:
- In my opinion, the use of the outcome variables should differ – to more relevant variables in the context of Pa and nutrition, e.g. health literacy.